# Shear Localization and Mechanical Properties of Cu/Ta Metallic Nanolayered Composites: A Molecular Dynamics Study

Chuanbin Wang [1], Junjie Wang [1], Jianian Hu [2], Shanglin Huang [1], Yi Sun [1], Youlin Zhu [1], Qiang Shen [1] and Guoqiang Luo [1,3,*]

[1] State Key Lab of Advanced Technology for Materials Synthesis and Processing, Wuhan University of Technology, Wuhan 430062, China; wangcb@whut.edu.cn (C.W.); 290687@whut.edu.cn (J.W.); 290818@whut.edu.cn (S.H.); sunyi@wuht.edu.cn (Y.S.); 250430@whut.edu.cn (Y.Z.); sqqf@whut.edu.cn (Q.S.)

[2] State Key Laboratory of Blasting Engineering, Jianghan University, Wuhan 430014, China; hjn@jhun.edu.cn

[3] Chaozhou Branch of Chemistry and Chemical Engineering Guangdong Laboratory, Hanjiang Laboratory, Chaozhou 521000, China

[*] Correspondence: luogq@whut.edu.cn

**Abstract:** With their excellent mechanical properties, Cu/Ta metallic nanolayered composites (MNCs) are extensively applied in aerospace and nuclear industry facilities. However, shear localization severely disrupts the ability of these materials to deform uniformly, attracting many researchers. The necessary time and length conditions of experiments limit the investigation of shear localization; thus, relevant studies are insufficient. The molecular dynamics simulation perfectly corresponds to the short duration and high strain rate of the deformation process. Therefore, in this study, we used molecular dynamics simulations to explore the effect of layer thickness on the shear localization of Cu/Ta MNCs with Kurdjumov–Sachs (KS) orientation–related interfaces. Our research demonstrates that shear localization occurs in samples with layer thicknesses below 2.5 nm, resulting in an inverse size effect on the flow strength. The quantitative analysis indicates that the asymmetry of dislocations in the slip transmission across the interface causes interface rotation. This activates dislocations parallel to the interface to glide beyond the distance of individual layer thicknesses, eventually forming shear bands. Both interface rotation and sliding dominate the plastic deformation in the shear band region. In addition, the dislocation density and amorphous phase increase with decreasing layer thickness.

**Keywords:** metallic nanolayered composites; shear localization; thickness dependence; interface

## 1. Introduction

Currently, Cu matrix metallic nanolayered composites (MNCs), with their excellent mechanical behaviors, are extensively applied in aerospace and nuclear industry facilities [1–3]. Recent research demonstrates, for example, that Cu matrix MNCs have become good candidates for replacing the outer layer of nuclear fusion ignition facility (NIF) capsules. This is due to their small grain-size, which produce smaller fragments during failure and cause much less damage to the expensive laser optics in the NIF target chamber [4].

However, the plastic instability of Cu matrix MNCs produced by their shear localization severely disrupts the ability of the material to deform uniformly, limiting its application [5–9]. To effectively control this instability, the process of shear bands (SBs) formation in Cu matrix MNCs needs to be explored. Several mechanisms have been presented to explain the formation of shear bands, such as dislocation transmission across interfaces [10,11], local lattice re-orientations [12,13], and localized rotation of interfaces [14,15]. Recent experimental studies have found that the formation of shear bands in Cu matrix MNCs exhibits a significant length–scale dependence [15–17]. For example, in indentation tests of Cu/Ta MNCs, the results show that the shear distance of grain boundary sliding or rotation, caused by layer buckling beyond the layer thickness, results in the formation of

shear bands [16]. This research on rolled Cu/Nb multilayers showed that significant plastic instability was observed at a 4 nm layer thickness, while the 40 nm samples could deform uniformly with no shear band formation [18]. It is difficult to observe experimentally, however, the role of layer thickness during formation and deformation behaviors in the shear band region.

With the development of computer simulation techniques, molecular dynamics simulations have become an important tool for studying both the deformation behavior and the mechanical properties of materials. These techniques have been successfully applied to study the evolution of defects and plastic deformation processes in shearing [19,20], nano-scratching [21], impact [22,23], tension, and compression [24,25]. Feng et al. explored the effect of the interface of crystalline Cu/amorphous $Cu_{50}Zr_{50}$ on the shear band using MD simulations. They found that introducing an amorphous interface effectively promoted the homogeneous distribution of the shear transition zones [26]. Sterwerf et al. investigated the shear band propagation of MNCs consisting of $Cu_{45}Zr_{55}$ amorphous Cu crystals under nanoindentation. Their results reveal that the deformation mode of the sample shifts from shear bands to co-deformation, with the thickness of the crystalline Cu layer increasing from 5 nm to 150 nm [27]. Vardanyan et al. analyzed the plasticity processes of the bilayer using a $Cu_{64.5}Zr_{35.5}$ amorphous Cu crystal to cut. MD simulations show that when a bilayer is cut parallel to the interface, the top crystal layer is deformed by dislocation control, while the top glass layer forms a series of shear bands [28].

The metal Ta exhibits many extraordinary mechanical properties, such as a high melting point, good ductility, excellent strength, and corrosion resistance [29]. In addition, Cu/Ta MNCs are widely used in semiconductor devices, so studying their deformation behavior and mechanical properties is valuable [30]. Lu et al. investigated the effect of a modulation period (7.4–22.36 nm) on the deformation behavior of Cu/Ta MNCs under uniaxial tension [31]. Tran further systematically explored the effects of temperature, strain rate, and modulation period on the phase transformation and fracture behavior of Cu/Ta MNCs under tension in the vertical and parallel interface directions [32]. However, the length–scale dependency of shear localization mentioned above has not been investigated in Cu/Ta MNCs.

In this study, the deformation behavior and mechanical properties of Cu/Ta MNCs with KS orientation-related interfaces were investigated. Molecular dynamics simulations of uniaxial compression were carried out for samples with layer thicknesses ranging from 1.25 nm to 16.8 nm, at 50 K and 300 K. This study found that the plastic instability of Cu/Ta MNCs has a significant length–scale dependency, with shear band formation and material softening observed in samples below the critical thickness. We expect to provide a valuable result for understanding the formation of plastic instability in Cu/Ta MNCs.

## 2. Computational Details

### 2.1. Interatomic Potential

A reasonable choice of interatomic potential is the basis for the accuracy of molecular dynamics simulation. Embedded atom potential (EAM) has been applied to many simulations such as interface, fracture, dislocation, diffusion, structural transformation, etc. EAM can provide reasonable stacking fault energy, surface energy, vacancy formation, migration energy, surface relaxation energy, and many other properties of metals [33,34]. The EAM potential function equation verified by Zhou et al. has been successfully performed to describe the microstructural characteristics of different nano-metal multilayer film systems. The EAM potential function proposed by Zhou et al. will also be executed to describe the interaction between Cu and Ta atoms in this simulation [35].

### 2.2. Molecular Dynamics Model

In this work, the initial box of Cu/Ta MNCs consists of 1.97 million atoms, and the cell size of the box is about $291 \times 283 \times 325$ Å$^3$. Periodic boundary conditions are applied in the three-axis direction of the box during the simulation. The interfacial plane orientation of

the Cu/Ta MNCs is $\{1\,1\,1\}_{Cu}\,||\,\{1\,1\,0\}_{Ta}$, and the specific orientation relations are shown in Table 1. This simulation is carried out by using the Large-scale Atomic/Molecular Massively Parallel Simulator (LAMMPS) code [36]. First, the conjugate gradient (CG) method is adopted to derive a minimum equilibrium energy model. The stopping tolerances for energy and force are $10^{-6}$ eV and $10^{-12}$ ev/Å, respectively. Then, the system is further balanced via a Nosé–Hoover thermostat [37]. During the relaxation period, the system uses isothermal–isobaric (NPT) ensembles, and the pressure in the three-axis directions is set to zero for 50 ps. Previous studies of face-centered cubic (FCC)/body-centered cubic (BCC) interface multilayers have shown that the behavior of defect evolution and deformation mechanisms in the fcc phase has not varied significantly in the strain rate range of $10^7$–$10^9$ 1/s [31]. Therefore, uniaxial compressive deformation was performed along the z-axis direction with a strain rate of $1.0 \times 10^9$ s$^{-1}$. The system fixes the pressure on the x- and y-axis direction to zero during the deformation process. The temperature during the simulation can be controlled by setting the temperature parameters of the Nosé–Hoover thermostat and NPT ensembles.

**Table 1.** Orientations and sizes of Cu/Ta MNCs with Kurdjumov–Sachs orientation–related interfaces.

| Interface | Constituent | Lx | Ly | Lz | Box Size |
|---|---|---|---|---|---|
| KS{1 1 1}<1 1 0>Cu\|\|{1 1 0}<1 1 1>Ta | Cu layer | 114 [0 1 −1] | 64 [−2 1 1] | [1 1 1] | 291 × 283 × 325 Å$^3$ |
| | Ta layer | 102 [−1 1 −1] | 35 [−2 −1 1] | [0 1 1] | |

Atomic data from MD simulations were post-processed by the visualization software OVITO, developed by Stukowski [38]. Dislocation Extraction Algorithm (DXA) can characterize the local structural environment and is used to identify the lattice types of crystals such as fcc, hcp, bcc, and others. This algorithm can also identify all dislocation line defects in atomic crystals, determining their Burgers vectors and dislocation lengths.

## 3. Results

Figure 1a shows the stress–strain curves for six thicknesses of the compressed samples at temperatures of 50 K and 300 K. The stress rises linearly with strain in the elastic phase until a peak occurs, after which it drops, and the sample enters the plastic deformation phase. The appearance of a stress peak prior to plastic flow has also been observed in other MD simulations [31], which is related to the high strain rates in the simulations. Compared to the defect-free simulated samples, the experimental samples contain dislocations prior to deformation. They are more prone to activation under stress, which results in the yield strength being lower than the peak strength of the simulated stress–strain curve. Therefore, we chose an average flow stress with a strain range of 7.5–15% to measure the model's strength. As shown in Figure 1b, the flow intensity of the sample peaks at a critical thickness ($h_C$) of 2.5 nm. For $h > h_C$, the strength of MNCs increases with decreasing layer thickness. Meanwhile, for $h < h_C$, the strain softening occurs. The simulated results are identical to those observed in the experiments. Furthermore, the results at both temperatures demonstrate that the compression strength decreases with an increase in temperature. As the temperature increases, the thermal motion of the atoms intensifies and the bonding of the atoms becomes weaker, which results in an increase in the proportion of amorphous phases [32] and a decrease in the critical stress for dislocation sliding. Therefore, the plastic deformation of the material at 300 K is easier to initiate, and the compressive strength is lower.

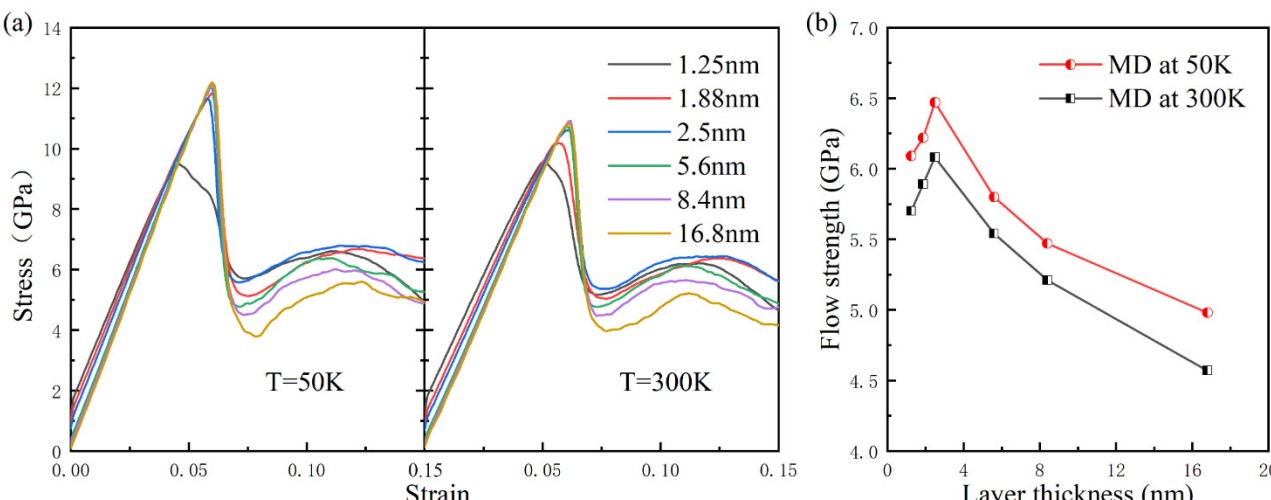

**Figure 1.** (**a**) Stress–strain curves for compressed samples with different layer thicknesses at 50 K and 300 K. (**b**) The average flow strength of the MD for samples of different layer thicknesses.

Two samples below and above $h_C$ are selected to analyze the plastic deformation mechanism of Cu/Ta MNCs, and their stress–strain curves are shown in Figure 2. Figure 3 displays snapshots of the defect evolution of the $h$ = 5.6 nm sample at four strain points of $A_1$–$D_1$. For a clear visualization, the atoms of the fcc and bcc lattice types are deleted, leaving only the hcp and other atoms. As the sample reaches the yield point (A1), Shockley partial dislocation nucleation can be observed on the Cu layer side, near the interface (green line in Figure 3). Nucleated dislocations subsequently spread as partial dislocation rings along the {1 1 1} slip plane, leaving extrinsic stacking faults (SFs). To relieve stress and accommodate strain, more Shockley partial dislocations nucleate from the interface and extend into the Cu layer, and 1/2<1 1 1> dislocations are generated on the Ta side of the layer (as shown in Figure 3b). It is possible to expect a rapid drop in stress after point $B_1$. As the stress drops to point $C_1$, multiple dislocation slip systems in the Cu layer are activated, forming intertwined dislocation forests and 1/6<1 1 0> stair-rod dislocations, blocking dislocation movements. Subsequently, the stress begins to rise to $D_1$, some dislocations in the Ta layer disappear, and stacking fault tetrahedra appear in the Cu layer. The weak interface (KS orientation relationship) is able to trap sliding dislocations, thus effectively hindering dislocation transmission across the interface [39]. As a result, the stresses in dislocation bending are less than the critical stresses for interface transmission, and dislocations are confined to slide within the layers [18], each layer forming relatively uniformly distributed dislocations. Thus, for $h \geq h_C$, the sample can deform uniformly, and no shear localization is observed.

Figure 4 presents snapshots of the defect evolution of the $h$ = 1.25 nm sample at four strain points of $A_2$–$D_2$. The sample yields at $\varepsilon$ = 0.051 (point $A_1$). Subsequently, dislocations preferentially nucleate at some interfaces and extend into the copper matrix, leaving stacking faults (red atoms). Owing to the thinness of the layer, dislocations are pinned by two adjacent interfaces. The movement and bending of pinned dislocations require high stresses, which are higher than the critical stress of a single dislocation ring crossing the interface [18]. The dislocation in the Cu layer can slip transmission directly across the interface, activating the 1/2<1 1 1> dislocation in the Ta layer. It is notable that interface rotation is observed in regions where local dislocation slip transmission across the interface occurs (the boxed area in Figure 4b). The strain increases to 0.118 (point $C_2$) and more regions undergo local interface rotation. Eventually at $\varepsilon$ = 0.14 (point $D_2$), shear bands form in the area of interface rotation.

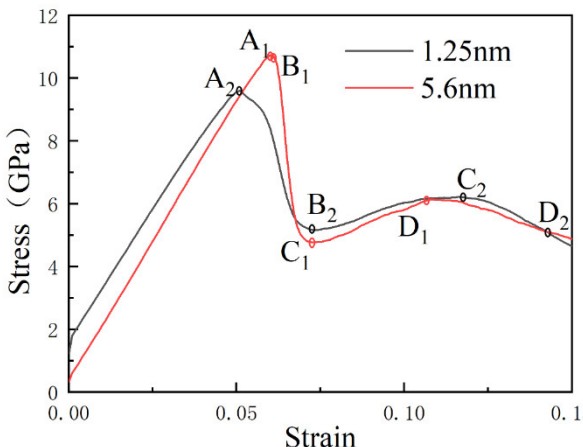

**Figure 2.** The stress–strain curves for compressed samples with *h* = 1.25 nm and *h* = 5.6 nm at 300 K. Points $A_1$–$D_1$ and $A_2$–$D_2$ are the four strain points of the stress-strain curve for the *h* = 5.6 nm and *h* = 1.25 nm samples respectively.

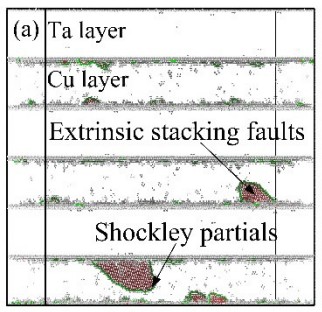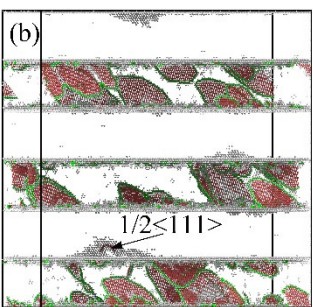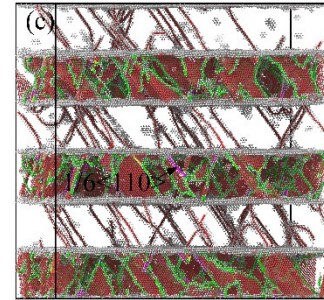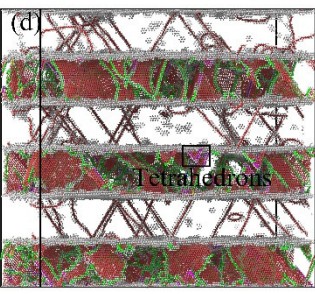

**Figure 3.** Snapshots of the defect evolution for the sample with *h* = 5.6 nm: (**a**) $\varepsilon = 0.059$ (point $A_1$), (**b**) $\varepsilon = 0.061$ (point $B_1$), (**c**) $\varepsilon = 0.073$ (point $C_1$), (**d**) $\varepsilon = 0.108$ (point $D_1$).

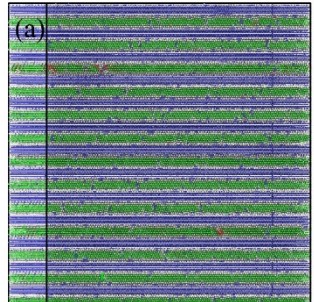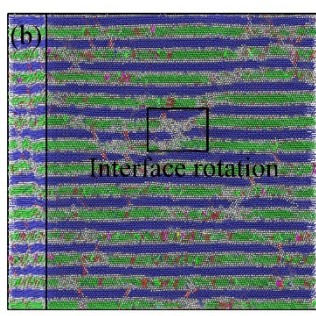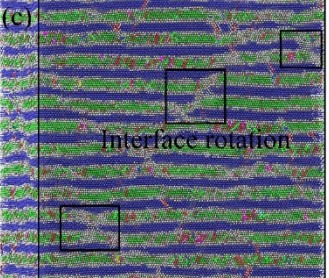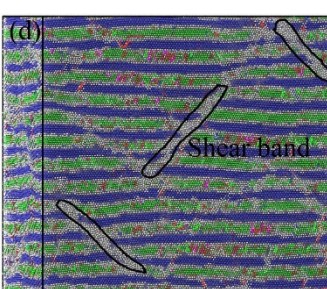

**Figure 4.** Snapshots of the defect evolution for the sample with *h* = 1.25 nm: (**a**) $\varepsilon = 0.051$ (point $A_2$), (**b**) $\varepsilon = 0.073$ (point $B_2$), (**c**) $\varepsilon = 0.118$ (point $C_2$), (**d**) $\varepsilon = 0.14$ (point $D_2$).

The shear localization of Cu/Ta MNCs involves two processes: interface rotation and shear band formation. Figure 5a(i–iv) illustrate the interface variations before and after dislocation slip transmission across the interface. At a strain of 0.06, the interface plane orientation of the pinned Cu layer dislocations remains unchanged. Subsequently, dislocation slip transmission across the interface takes place at $\varepsilon = 0.065$. For the KS orientation relation interface, three $\{1\ 1\ 1\}_{Cu}$ and five $\{1\ 1\ 0\}_{Ta}$ slip planes intersect it, but only one set of intersecting trajectories is parallel [39], as shown in Figure 5c. As the incoming dislocations approach the interface, the dislocations are relatively easy to transport along the $<-1\ 1\ -1>_{Cu}$ and $<-0\ 1\ -1>_{Ta}$ intersecting trajectories, due to the low transport energy of the parallel trajectories. A similar phenomenon has been observed in the Cu/Nb system [40]. Due to the asymmetry of dislocation slip transmission across

the interface, local interface rotation occurs at a strain of 0.075 (Figure 5a(iii)). The similar dislocation transmission events take place in adjacent layers, eventually generating highly locally defected structures (the boxed region in Figure 5a(iv)). Note that the Ta layer becomes thinner, the Cu layer becomes thicker, and so the layer thickness becomes uneven. Figure 5b provides snapshots of the atomic misorientation prior to and after dislocation transmission events. The initial interface separates the Cu and Ta layers with different atomic orientations, and each interface is parallel to each other. The atomic orientation of the unyielding Ta layer is uniform, while the atomic orientation of the yielding Cu layer is partially changed. After local interface rotation occurs, the atoms reorient, the thickness of the matrix becomes inhomogeneous, and highly localized structures appear.

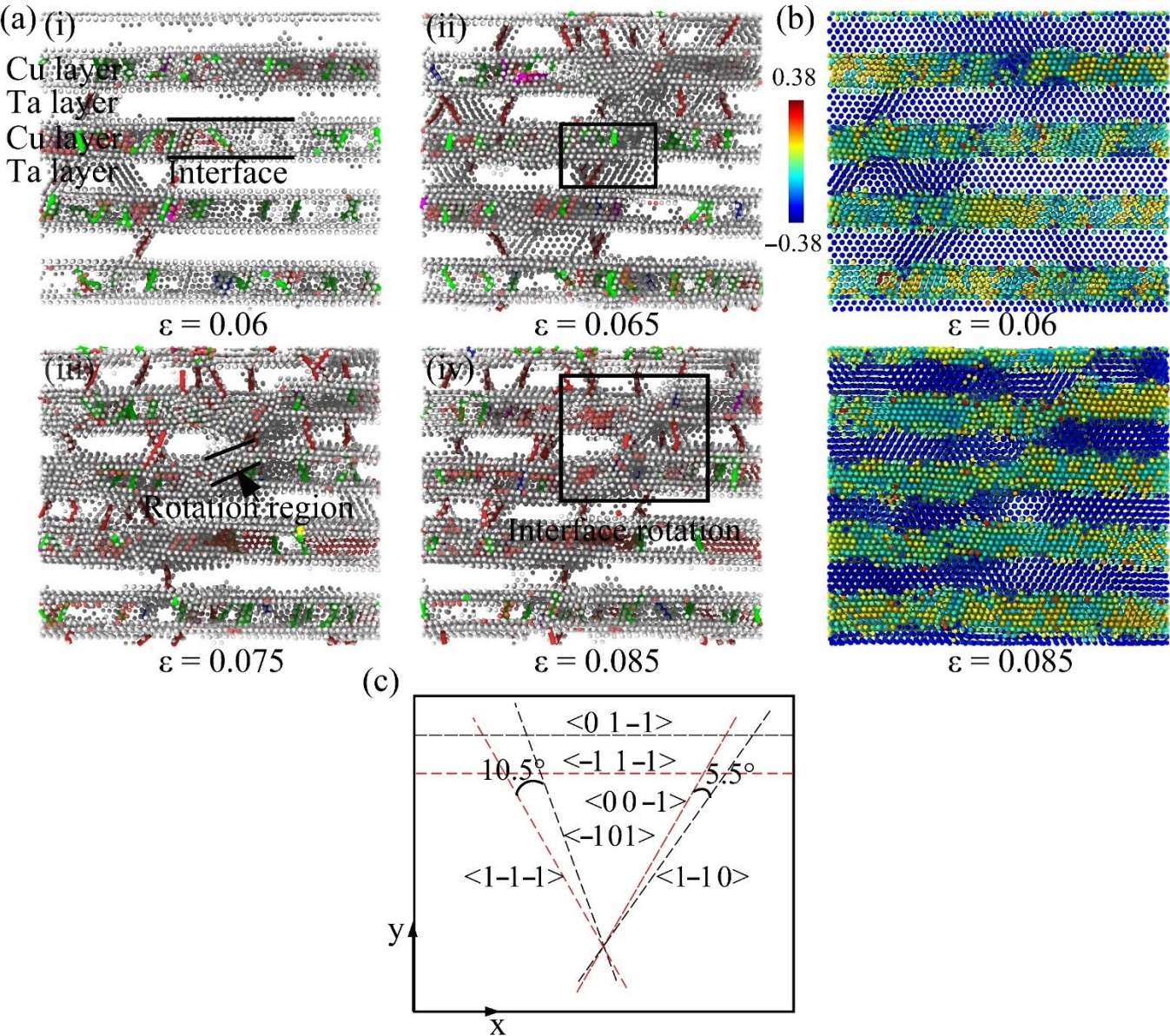

**Figure 5.** (**a**) Snapshots of the local atomic defect evolution for interface rotation. Images (**i–iv**) correspond to applied strains at 0.06, 0.065, 0.075, 0.085, respectively. (**b**) Misorientation and (**c**) trajectories of the intersection of the $\{1\,1\,1\}_{Cu}$ and $\{1\,1\,0\}_{Ta}$ slip planes with the KS orientation-related interface. Three black and red dashed lines represent the intersection of the interface plane with the $\{1\,1\,1\}_{Cu}$ and $\{1\,1\,0\}_{Ta}$ slip planes, respectively.

Figure 6a,b exhibit snapshots of the atomic defects of the shear band formation. The initial Schmidt factor for the $(1-1\ 1)_{Cu}$ and $(-1\ 0\ 1)_{Ta}$ interface plane is zero, since the applied load is perpendicular to the interface when the sample is compressed along the z-axis. As the local interface rotation proceeds, the Schmidt factor increases. The interface plane of $(1-1\ 1)_{Cu}$ and $(-1\ 0\ 1)_{Ta}$ generates shear stresses, activating the sliding dislocations in that plane. The activated dislocations glide beyond the distance of individual layer thicknesses, eventually forming the shear band (as shown in Figure 6b). Figure 6c demonstrates a snapshot of the dislocation line after the shear band formation. The dislocations in the shear band region disappear, revealing that their plastic strain is dominated by interface rotation and sliding, rather than dislocation sliding.

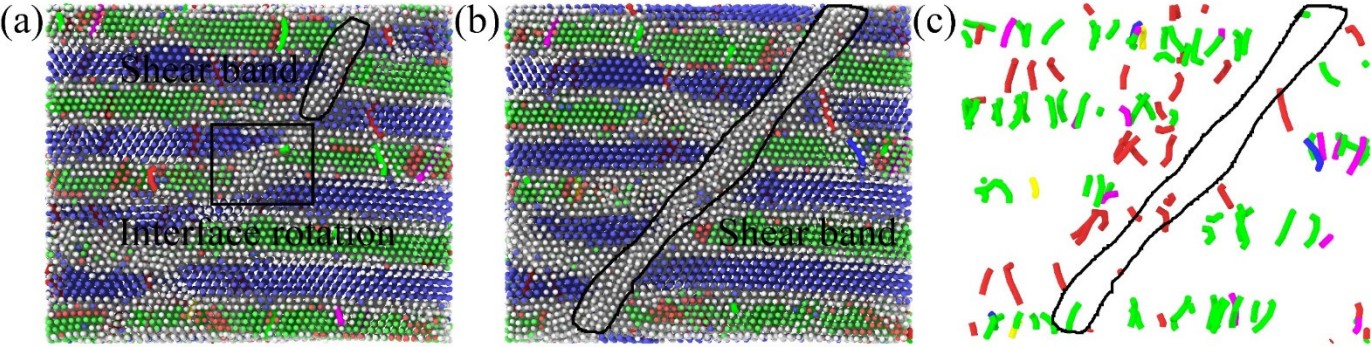

**Figure 6.** Snapshots of the local atomic defect evolution in the shear band formation. (**a**) $\varepsilon = 0.13$, (**b**) $\varepsilon = 0.15$, (**c**) dislocation lines at $\varepsilon = 0.15$. The rectangular area undergoes interface rotation and the ribbon area forms shear bands.

Figure 7a illustrates snapshots of the von Mises shear strain distribution for the $h = 1.25$ nm sample at different strain levels. The definition of von Mises shear strain has been described in [41]. It can be observed that the early shear strain is concentrated near the interface and is mainly attributed to dislocation nucleation at the interface and slip transmission across the interface. As the strain increases, dislocations shear along the rotating interface, resulting in high shear strains at local areas within the layer. At $\varepsilon = 0.15$, the early shear localization region thickens and intensifies, forming narrow shear bands. The change in atomic configuration in Figure 7b also reveals that shear band formation disrupts the ability of the sample to deform uniformly.

To reveal more details of the plastic deformation mechanism of Cu/Ta MNCs below and above $h_C$, Figure 7c,d present the radial distribution functions (RDF) of the $h = 1.25$ nm and $h = 5.6$ nm samples at 300 K, respectively. The radial distribution function (RDF) is commonly applied to investigate the internal ordering of the substance. The crystal possesses a regular periodic structure, and the corresponding RDF curve exhibits long-range peaks. The RDF curve for $h = 5.6$ nm still possesses long-range peaks after deformation, but the peak intensity decreases. This indicates that the internal crystal structure is not transformed, and the dislocation-slip-dominated plastic deformation mode reduces the structured orderliness of the sample. However, compared to the two strain points of 0 and 0.15, several peak points of the RDF curve for $h = 5.6$ nm decline in value noticeably lower than for $h = 1.25$ nm. Furthermore, it is worth noting that the long-range ordered peak of the RDF curve for the $h = 1.25$ nm sample disappears. The disappearance of the long-range peaks is mainly attributed to the disruption of the internal crystal structure caused by the shear band generated due to interface rotation and sliding. Thus, interface rotation and sliding dominate the plastic deformation in the shear band region.

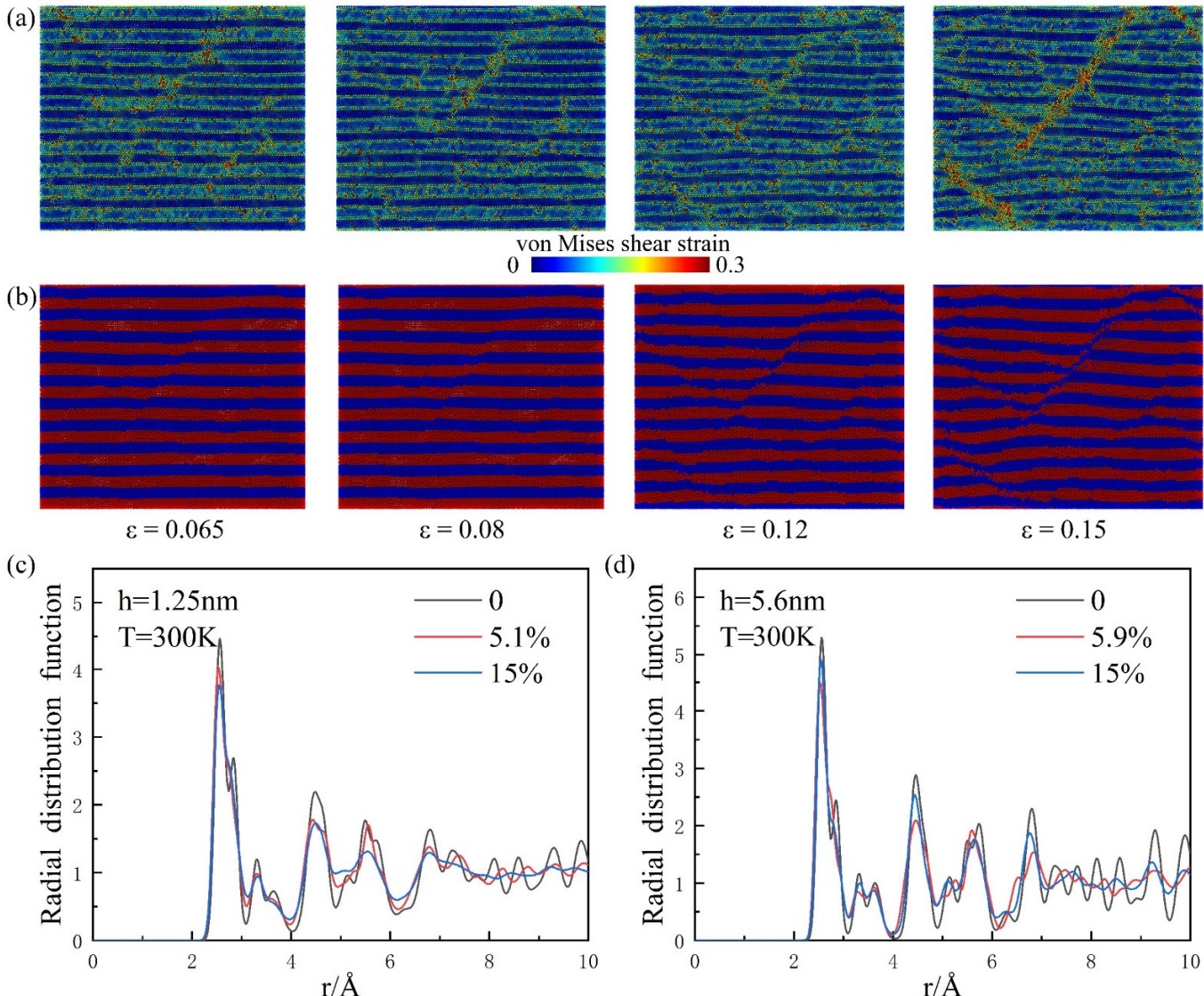

**Figure 7.** Snapshots of (**a**) von Mises shear strain distribution and (**b**) atomic configuration of
*h* = 1.25 nm samples at different strain values. The radial distribution function for (**c**) *h* = 1.25 nm and
(**d**) *h* = 5.6 nm samples at 300 K.

Figure 8 displays the dislocation density evolution curves with strain during uniaxial
compression for samples with different layer thicknesses at 50 K and 300 K. The dislocation
density evolution is roughly divided into three stages. The first is the elastic deformation
stage, in which the dislocation density is almost zero. The second is the yielding stage,
when the samples yield, the dislocations nucleate at the interface and expand in the matrix,
and the dislocation density rises rapidly to its first peak. The third stage is the plastic
deformation stage, in which dislocations close to the interface are trapped and the fcc and
bcc phases are transformed into amorphous phases [32], all of which lead to a decrease
in dislocation density. After that, the proliferation and obliteration of dislocations in the
matrix tend to be dynamically balanced, and the dislocation density turns stable. It is
found that the dislocation density increases as the layer thickness decreases. A rational
explanation for this is that the number of interfaces is greater at smaller layer thicknesses,
which can provide more nucleation nodes for dislocations. However, for the sample with
*h* = 1.25 nm, the dislocation density decreases significantly in the third stage, owing mainly
to the replacement of previous dislocations by amorphous defects in the shear band region.
This is consistent with the tension studies by Tran et al. in their study of dislocation density

in Cu/Ta nanostructures. It is because more fcc and bcc phases transform amorphous phases at higher temperatures [32].

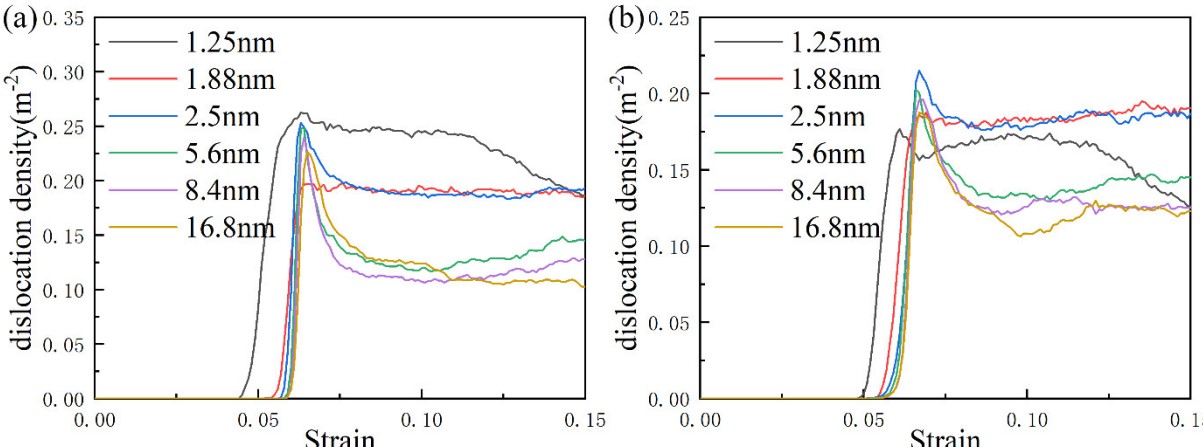

**Figure 8.** The dislocation density evolution curves during uniaxial compression: (**a**) 50 K, (**b**) 300 K.

To reveal the details of the phase transition of Cu/Ta MNCs during compression below and above $h_C$, Figure 9a,b exhibit the phase transition evolution curves for $h = 1.25$ nm and $h = 5.6$ nm samples at 300 K, respectively. For the $h = 1.25$ nm sample, the face-centered cubic structure transforms to the hcp phase at $\varepsilon = 0.05$, while the face-centered cubic and body-centered cubic structures transform to the amorphous phase. For the $h = 5.6$ nm sample, the same phase transition is observed at $\varepsilon = 0.06$. However, during the strain of 0.075–0.15, more fcc and bcc phases transform to amorphous phases for $h = 1.25$ nm, as a result of amorphous shear bands, while the proportion of various atomic structures for $h = 5.6$ nm remains approximately constant.

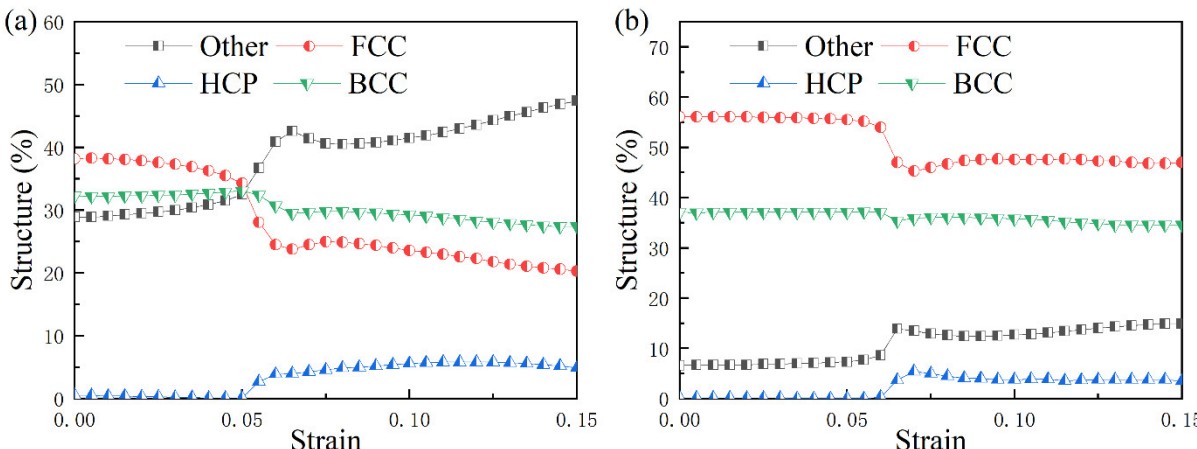

**Figure 9.** Atomic percentages of bcc, fcc, hcp and other disordered structures during uniaxial compression at 300 K: (**a**) $h = 1.25$ nm, (**b**) $h = 5.6$ nm.

## 4. Conclusions

Molecular dynamics simulations were performed to investigate the thickness dependence of the shear localization and mechanical properties of Cu/Ta MNCs. The evolution of defects at the atomic level is investigated to reveal the process of shear band formation and plastic deformation behavior. The conclusions are as follows:

(1)　The compressive strength of Cu/Ta MNCs increases with decreasing thickness up to a critical thickness of 2.5 nm, below which softening begins. This is attributed to the

formation of shear localization. Moreover, the compressive strength increases with decreasing temperature;

(2) The geometry analysis of the slip system and the evolution of defects at the atomic level indicate that asymmetry of dislocation transmission across interfaces causes interface rotation, activating dislocations parallel to the interface to glide beyond the distance of individual layer thicknesses, and eventually forming shear bands;

(3) The plastic deformation above $h_C$ is dominated by dislocation sliding, while the plastic strain in the shear band region below $h_C$ is dominated by interfacial rotation and sliding;

(4) The dislocation density is higher with lower layer thickness and temperature, but the shear band formation reduces dislocation density;

(5) The transformation of local lattice regions into amorphous shear band regions results in more face-centered cubic and body-centered cubic structures transformed into amorphous phases for samples below $h_C$.

**Author Contributions:** Conceptualization, C.W. and J.W.; Data curation, J.W.; Funding acquisition, Q.S.; Methodology, J.W.; Resources, G.L.; Software, J.H. and Y.Z.; Supervision, G.L. and Y.S.; Visualization, S.H.; Writing (original draft), J.W.; Writing (review and editing), G.L. All authors have read and agreed to the published version of the manuscript.

**Funding:** Key-Area Research and Development Program of Guangdong Province (Grant No. 2020 B010181001).

**Institutional Review Board Statement:** Not applicable.

**Informed Consent Statement:** Not applicable.

**Data Availability Statement:** The data generated and analyzed in this study are not publicly available for legal/ethical reasons, but are available from the corresponding authors upon reasonable request.

**Acknowledgments:** The authors gratefully acknowledge the financial supports from Key-Area Research and Development Program of Guangdong Province (Grant No. 2020B010181001). This study was also supported by Guangdong Major Project of Basic and Applied Basic Research (Grant No. 2021B0301030001), National Key R&D Program of China (2021YFB3802300), National Natural Science Foundation (51932006), and Basic strengthening project of science and Technology Commission of CMC (202022JQ01).

**Conflicts of Interest:** The authors declare no conflict of interest.

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
