# Peer review of "Shear Localization and Mechanical Properties of Cu/Ta Metallic Nanolayered Composites: A Molecular Dynamics Study"

_metals, doi:10.3390/met12030421_

Round 1

Reviewer 1 Report

The manuscript is well organized and illustrates the scientific content of the work.
This paper is publishable after some minor revisions.
Mistake: Page 2, line 62, “Vardan et al. analyzed ..............” In References, it is written: [28] Vardanyan..

Lines 96-98: Explain why you chose these values. Are these ones optimal?

Minor grammar mistakes: lines: 173, 216, 217.   

Lines 134-136, 202-203, 239-240, 255-258: Insert more details.

Minor mistakes in References: [38].

Specify deeper the advantages and disadvantages of the proposed molecular dynamics study. Also, specify the limits of this study.

Insert Author Contributions, Data Availability Statement, Conflicts of Interest.

If possible, I recommend to be cited the following reference:

[1] Applied Sciences, vol. 12, no. 3, article 946, p. 1-15, 2022. DOI: 10.3390/app12030946.

This paper presents an interesting approach and deserved to be published after the mentioned revisions.

Author Response

I am honored to receive your suggestions, which will help a lot in revising the article. The following is a response to your suggestion.
1. The grammar of the article, the name of the author of reference [28] and the format of [38] were modified.
2.The explanation of the choice of values in rows 96-98 is as follows. Molecular dynamics simulation needs to build a reasonable size box, the larger the box size, the more accurate the simulation will be. However, the corresponding simulation time will increase a lot. Therefore, the box for this simulation is chosen to be approximately 30 nm long in the three axes to ensure the accuracy of the simulation, while making the Cu and Ta layers coincide in each axis direction, so these values are chosen.

3.For lines 134-136, 202-203, 239-240, 255-258, insert more details (in the attachment)

4.The advantage of molecular dynamics studies is the exploration of defect evolution at the atomic scale, the disadvantage is the inability to provide studies with large size (microns or even larger) models, which is the limitation of this study.

5.Author contributions, data availability statement, conflicts of interest have been inserted in the article.Please see the attached files.

6.The references you provided provide very interesting approaches and deserve to be cited. However, I was not able to find the corresponding article and will cite it when I do.

Looking forward to your reply!

Reviewer 2 Report

Please find my comments below:

  1. Was the choice of the two thicknesses to analyze in figure 2 based on that they exhibit similar flow stresses?
  2. In figure 3, which layer is the Cu and which the Ta?
  3. Is the increase of stress between Cu and Ta caused by the trapping of dislocations at the interface, thus preventing them from crossing?
  4. At which strain does the first dislocation cross from the Cu to Ta layer? When this occurs, a drop in stress should be expected. Is this happening at point B1? A first dislocation seems to cross at that point, which could explain this, but it is not clear from the text.
  5. For the smaller thickness case, why the change in stress between C and D points is the same as in the thicker layer? If the dislocations are pinned, shouldn't this change be more significant?

Author Response

I am honored to receive your suggestions, which will help a lot in revising the article. The following is the response to your suggestion.

1.The two thicknesses do exhibit similar flow stresses and the authors wish to explore the mechanisms of plastic deformation for samples above and below the critical thicknesses, which are well explained by the two thicknesses represented in Figure 2.

2.Sorry for the lack of annotation in the figure, the annotation will be added to the image in the attached text. The Shockley part of the dislocation was first generated as a Cu layer, and the 1/2<111> dislocation was generated as a Ta layer.

3.The weak interface (KS orientation relationship) is able to trap sliding dislocations and therefore effectively hinders the transport of dislocations across the interface. In addition, the decrease in dislocation density, as well as interactions between dislocations (step-rod dislocations and dislocation forests) all lead to an increase in stress between Cu and Ta.

4.At a strain of 6.1% (point B1), 1/2<111> dislocations are generated near one side of the Ta layer and the Ta layer begins to yield, when a drop in stress can be expected. The text in the appendix has been added.

5.In the case of smaller thicknesses, dislocations are pinned to the interface on both sides. The stresses follow the interface barrier strength (IBS) for the transmission of a single glide dislocation.The formation of shear bands causes a more pronounced drop in stress between points C2 and D2 of the material. In the case of larger thicknesses, the interface blocks dislocation transport and the stresses follow the confined layer slip model. the stresses between points C1 and D1 are mainly influenced by dislocation-dislocation interactions and dislocation-interface interactions.

Thank you for your guidance on this article once again!

Reviewer 3 Report

The authors investigated the deformation behaviour and mechanical properties of Cu/Ta MNCs  with KS orientation-related  interfaces are investigated. Molecular dynamics simulations  of uniaxial compression are  carried out for samples with layer thicknesses ranging from  1.25 nm to 16.8 nm  at 50 K and 300 K. It is found that the plastic instability of Cu/Ta MNCs  has a significant length-scale dependency,  with shear band formation and material softening observed in samples below the critical thickness. We expect to provide a valuable result for understanding the formation of plastic instability in Cu/Ta MNCs.

The manuscript could be accepted after major revision.

The literature should be supported by published articles in metals.

The abstract and introduction should be improved.

Why does the plastic deformation above  hC  is dominated by dislocation sliding?

Use line markers with different styles in Fig 1.

Why there are more face-centred cubic and body-centred cubic structures transformed  296 into amorphous phases below hC.

The discussions should be improved.

Why does dislocation interactions in  the matrix cause the dislocation density to first decrease and then stabilise.

Author Response

I am honored to receive your suggestions, which will help a lot in revising the article. The following is the response to your suggestion.

1.The article investigates Cu/Ta metal nanolayered composites and therefore cites articles on Cu/Nb, Cu/Zr,Cu/Au systems and published articles on Cu/Ta systems as the basis. For example, references [7-12][29-31], [39-41]...

2.The annexed text provides some improvements to the abstract and introduction.

3.The radial distribution function(RDF) is commonly applied to investigate the internal ordering of the substance. The crystal possesses a regular periodic structure, and the corresponding RDF curve exhibits a long-range peak. The RDF curve above hc still posses long-range peaks after deformation, but the peak intensity decreases. This indicates that the internal crystal structure is not transformed and the dislocation-slip-dominated plastic deformation mode reduces the structure orderliness of the sample.

4.Figure 1(b) shows the average flow intensity curves for samples with different layer thicknesses. If the same lines as in Fig. 1(a) are used, some points cannot be clearly observed, so the dotted line curve style is adopted.

5.The transformation of local lattice regions into amorphous shear band regions results in more face-centered cubic and body-centered cubic structures transformed into amorphous phases for samples below hc.

6.The text in the Appendix improves on the discussion.

7.I am sorry for the uncertainty in the meaning of the text expression. During the plastic deformation phase, dislocations close to the interface are trapped and the FCC and BCC phases are transformed into amorphous phases [32], which leads to a decrease in dislocation density. After that the proliferation and obliteration of dislocations in the matrix tend to be dynamically balanced and the dislocation density becomes stable.

Thank you for your guidance on this article once again!

Round 2

Reviewer 2 Report

The authors answered all my comments

Reviewer 3 Report

 Accept in present form.